# FEDERATED ENSEMBLE-DIRECTED OFFLINE REINFORCEMENT LEARNING

## ABSTRACT

We consider the problem of federated offline reinforcement learning (RL), a scenario under which distributed learning agents must collaboratively learn a high-quality control policy only using small pre-collected datasets generated according to different unknown behavior policies. Naïvely combining a standard offline RL approach with a standard federated learning approach to solve this problem can lead to poorly performing policies. In response, we develop the Federated Ensemble-Directed Offline Reinforcement Learning Algorithm (FEDORA), which distills the collective wisdom of the clients using an ensemble learning approach. We develop the FEDORA codebase to utilize distributed compute resources on a federated learning platform. We show that FEDORA significantly outperforms other approaches, including offline RL over the combined data pool, in various complex continuous control environments and real-world datasets. Finally, we demonstrate the performance of FEDORA in the real-world on a mobile robot.

## 1 INTRODUCTION

Federated learning is an approach wherein clients learn collaboratively by sharing their locally trained models (not their data) with a federating agent, which periodically combines their models and returns the federated model to the clients for further refinement (Kairouz et al., 2021; Wang et al., 2021). Federated learning has seen recent success in supervised learning applications due to its ability to generate well-trained models using small amounts of data at each client, while preserving privacy and reducing the usage of communication resources. There has also been interest in federated learning for *online* reinforcement learning (RL), wherein clients learn via sequential interactions with their environments and federating learned policies across clients (Khodadadian et al., 2022; Nadiger et al., 2019; Qi et al., 2021). However, such online interactions with real-world systems are often infeasible, and each client might only posses pre-collected operational data generated according to a client-specific behavior policy. The fundamental problem of federated *offline* RL is on how to learn the optimal policy only using such offline data collected by heterogeneous policies at clients, without actually sharing any of the data.

Offline RL algorithms (Levine et al., 2020), such as CQL Kumar et al. (2020b) and TD3-BC (Fujimoto & Gu, 2021) offer an actor-critic learning approach that only utilizes existing datasets at each client. However, in our case, this approach taken across many small datasets at clients will produce an ensemble of policies of heterogeneous (unknown) qualities across the clients, along with their corresponding critics of variable accuracy. We will see that naïvely federating such offline RL trained policies and critics using a standard federation approach, such as FedAvg (McMahan et al., 2017) can lead to a policy that is even worse than the constituent policies. We hence identify the following basic challenges of federated offline RL: (i) *Ensemble heterogeneity:* Heterogeneous client datasets will generate policies of different performance levels. It is vital to capture the collective wisdom of this ensemble of policies, not average them. (ii) *Pessimistic value computation:* Offline RL employs a pessimistic approach toward computing the value of actions poorly represented in the data to minimize distribution shift (and so reduce the probability of taking these actions). However, federation must be ambitious in extracting the highest values as represented in the ensemble of critics (and so promote high-value actions). (iii) *Data heterogeneity:* As with other federated learning, multiple local gradient steps based on heterogeneous data at each client between federation rounds may lead to biased models. We must regularize local policies to reduce such drift.

In this work, we propose Federated Ensemble-Directed Offline RL Algorithm (FEDORA), which collaboratively produces a high-quality control policy and critic function. FEDORA estimates the performance of client policies using only local data (of unknown quality) and, at each round of federation, produces a weighted combination of the constituent policies that maximizes the overall objective, while regularizing by the entropy of the weights. The same approach is followed to federate client critics. Following the principle of maximum entropy in this manner produces both federated policies and critics that extract the collective wisdom of the ensemble. In doing so, it constructs a federated policy and a critic based on the relative merits of each client policy in an ensemble learning manner. FEDORA ensures optimism across evaluation by the federated and local critic at each client and so sets ambitious targets to train against. It addresses data heterogeneity by regularizing client policies with respect to both the federated policy and the local dataset. Finally, FEDORA prunes the influence of irrelevant data by decaying the reliance on a dataset based on the quality of the policy it can generate. To the best of our knowledge, no other work systematically identifies these fundamental challenges of offline federated RL, or designs methods to explicitly tackle each of them.

We develop a framework for implementing FEDORA either on a single system or over distributed compute resources. We evaluate FEDORA on a variety of MuJoCo environments and real-world datasets and show that it outperforms several other approaches, including performing offline RL on a pooled dataset. We also demonstrate FEDORA's excellent performance via real-world experiments on a TurtleBot robot (Amsters & Slaets, 2020). **We provide our codebase, several experimental results and a video of the robot experiments in the supplementary material.**

## 2 RELATED WORK

**Offline RL:**The goal of offline RL is to learn a policy from a fixed dataset generated by a behavior policy (Levine et al., 2020). One of the key challenges of the offline RL approach is the distribution shift problem where the state-action visitation distribution of learned policy may be different from that of the behavior policy which generated the offline data. It is known that this distribution shift may lead to poor performance of the learned policy (Levine et al., 2020). A common method used by offline RL algorithms to tackle this problem is to learn a policy that is close to the behavior policy that generated the data via regularization either on the actor or critic (Fujimoto & Gu, 2021; Fujimoto et al., 2019; Kumar et al., 2020a; 2019; Wu et al., 2019). Some offline RL algorithms perform weighted versions of behavior cloning or imitation learning on either the whole or subset of the dataset (Wang et al., 2018; Peng et al., 2019; Chen et al., 2020). Yue et al. (2022; 2023) propose data rebalancing methods designed to prioritize highly-rewarding transitions that can be augmented to offline RL algorithms to alleviate the distribution shift issue for heterogeneous data settings.

**Federated Learning:** McMahan et al. (2017) introduced FedAvg, a federation strategy where clients collaboratively learn a joint model without sharing data. A generalized version of FedAvg was presented in Reddi et al. (2021). A key problem in federated learning is data heterogeneity wherein clients have non-identically distributed data, which causes unstable and slow convergence (Wang et al., 2021; Karimireddy et al., 2020; Li et al., 2020). To tackle the issue of data heterogeneity, Li et al. (2020) proposed FedProx, a variant of FedAvg, where a proximal term is introduced reduce deviation by the local model from the server model.

**Federated Reinforcement Learning:** Federated learning has recently been extended to the online RL setting. Khodadadian et al. (2022) analyzed the performance of federated tabular Q-learning. Qi et al. (2021) combined traditional online RL algorithms with FedAvg for multiple applications. Some works propose methods to vary the weighting scheme of FedAvg according to performance metrics such as the length of a rally in the game of Pong (Nadiger et al., 2019) or average return in the past 10 training episodes (Lim et al., 2021) to achieve better performance or personalization. Wang et al. (2020) proposed a method to compute weights using attention over performance metrics of clients such as average reward, average loss, and hit rate for an edge caching application. Hebert et al. (2022) used a transformer encoder to learn contextual relationships between agents in the online RL setting. Hu et al. (2021) proposed an alternative approach to federation where reward shaping is used to share information among clients. Xie & Song (2023) proposed a KL divergence-based regularization between the local and global policy to address the issue of data heterogeneity in an online RL setting.

In the offline RL setting, Zhou et al. (2022) propose federated dynamic treatment regime algorithm by formulating offline federated learning using a multi-site MDP model constructed using linear

MDPs. However, this approach relies on running the local training to completion followed by just one step of federated averaging. Unlike this work, our method does not assume linear MDPs, which is a limiting assumption in many real-world problems. Moreover, we use the standard federated learning philosophy of periodic federation followed by multiple local updates. *To the best of our knowledge, ours is the first work to propose a general federated offline RL algorithm for clients with heterogeneous data.*

## 3 PRELIMINARIES

**Federated Learning:** The goal of federated learning is to minimize the following objective,

$$F(\theta) = \mathbb{E}_{i \sim \mathcal{P}} \left[ F_i(\theta) \right], \tag{1}$$

where $\theta$ represents the parameter of the federated (server) model, $F_i$ denotes the local objective function of client $i$, and $\mathcal{P}$ is the distribution over the set of clients $\mathcal{N}$. The FedAvg algorithm (McMahan et al., 2017) is a popular method to solve Eq. (1) in a federated way. FedAvg divides the training process into rounds, where at the beginning of each round $t$, the server broadcasts its current model $\theta^t$ to all the clients, and each client initializes its current local model to the current server model. Clients perform multiple local updates on their own dataset $\mathcal{D}_i$ to obtain an updated local model $\theta_i^t$. The server then averages these local models proportional to the size of their local dataset to obtain the server model $\theta^{t+1}$ for the next round of federation, as

$$\theta^{t+1} = \sum_{i=1}^{|\mathcal{N}|} w_i \theta_i^t, \quad w_i = \frac{|\mathcal{D}_i|}{|\mathcal{D}|}, \quad |\mathcal{D}| = \sum_{i=1}^{|\mathcal{N}|} |\mathcal{D}_i|. \tag{2}$$

**Reinforcement Learning:** We model RL using the Markov Decision Process (MDP) framework denoted as a tuple $(\mathcal{S}, \mathcal{A}, R, P, \gamma, \mu)$, where $\mathcal{S}$ is the state space, $\mathcal{A}$ is the action space, $R : \mathcal{S} \times \mathcal{A} \to \mathbb{R}$ is the reward function, and $P : \mathcal{S} \times \mathcal{A} \times \mathcal{S} \to [0, 1]$ denotes the transition probability function that gives the probability of transitioning to a state $s'$ by taking action $a$ in state $s$, $\gamma$ is the discount factor, and $\mu$ is the distribution of the initial state $s_0$. A policy $\pi$ is a function that maps states to actions (deterministic policy) or states to a distribution over actions (stochastic policy). The goal of RL is to maximize the infinite horizon discounted reward of policy $\pi$, defined as $J(\pi) = \mathbb{E}_{\pi, P, \mu} \left[ \sum_{t=0}^{\infty} \gamma^t R(s_t, a_t) \right]$, which is the expected cumulative discounted reward obtained by executing policy $\pi$. The state-action value function (or Q function) of a policy $\pi$ at state $s$ and executing action $a$ is the expected cumulative discounted reward obtained by taking action $a$ in state $s$ and following policy $\pi$ thereafter: $Q^\pi(s, a) = \mathbb{E}_{\pi, P} \left[ \sum_{t=0}^{\infty} \gamma^t R(s_t, a_t) | s_0 = s, a_0 = a \right]$.

**Offline Reinforcement Learning:** The goal of offline RL is to learn a policy $\pi$ only using a static dataset $\mathcal{D}$ of transitions $(s, a, r, s')$ collected using a behavior policy $\pi_b$ without any additional interactions with the environment. Offline RL algorithms typically utilize some kind of regularization with respect to the behavior policy to ensure that the learned policy does not deviate from the behavior policy. This regularization is done to prevent distribution shift, a significant problem in offline RL, where the difference between the learned policy and behavior policy can lead to erroneous Q-value estimation of state-action pairs not seen in the dataset (Kumar et al., 2020a; Levine et al., 2020).

Our approach is compatible with most offline RL algorithms, such as CQL Kumar et al. (2020b) or TD3-BC Fujimoto & Gu (2021). We choose TD3-BC for illustration, motivated by its simplicity and its superior empirical performance in benchmark problems. The TD3-BC algorithm is a behavior cloning (BC) regularized version of the TD3 algorithm (Fujimoto et al., 2018). The policy in TD3-BC is updated using a linear combination of TD3 objective and behavior cloning loss, where the TD3 objective ensures policy improvement and the BC loss prevents distribution shift. More precisely, the TD3-BC objective can be written as

$$\pi = \arg\max_{\pi} U_{\mathcal{D}}(\pi), \text{where} \quad U_{\mathcal{D}}(\pi) = \mathbb{E}_{s, a \sim \mathcal{D}} \left[ \lambda Q^\pi(s, \pi(s)) - (\pi(s) - a)^2 \right], \tag{3}$$

and $\lambda$ is a hyperparameter that determines the relative weight of the BC term.

## 4 FEDERATED OFFLINE REINFORCEMENT LEARNING

In real-world offline RL applications, data is typically obtained from the operational policies of multiple agents (clients) with different (unknown) levels of expertise. Clients often prefer not to

share data. We aim to learn the optimal policy for the underlying RL problem using only such offline data, without the clients knowing the quality of their data, or sharing it with one another or the server. Furthermore, neither the clients nor server have access to the underlying model or the environment. We denote the set of clients as $\mathcal{N}$. Each client $i \in \mathcal{N}$ has the offline dataset $\mathcal{D}_i = \{(s_j, a_j, r_j, s'_j)_{j=1}^{m_i}\}$ generated according to a behavior policy $\pi_i^b$. We assume that the underlying MDP model $P$ and reward function $R(\cdot, \cdot)$ are identical for all the clients, and the statistical differences between the offline datasets $\mathcal{D}_i$ are only due to the difference in behavior policies $\pi_i^b$ used for collecting the data.

In a standard federated learning algorithm such as FedAvg, each client performs multiple parameter updates before sending its parameters to the server. It is known that performing multiple local updates in federated learning can reduce the communication cost significantly without compromising on the optimality of the converged solution (Kairouz et al., 2021; Wang et al., 2021). In federated offline RL, since each client has to perform multiple steps of policy evaluation and policy update using its local offline data $\mathcal{D}_i$, it is reasonable to consider a client objective function that is consistent with a standard offline RL algorithm objective. We choose the objective function used in the TD3-BC algorithm (Fujimoto & Gu, 2021), i.e., $U_{\mathcal{D}_i}$ given in Eq. (3), as the client objective function. Our choice is motivated by the simplicity of the TD3-BC objective function and its empirical success in a variety of environments. Similar to the standard federated learning objective given in Eq. (1), we can now define the federated offline RL objective as

$$U(\pi_{\text{fed}}) = \sum_{i=1}^{|\mathcal{N}|} w_i U_{\mathcal{D}_i}(\pi_{\text{fed}}), \tag{4}$$

where $w_i$ are weights to be determined.

One approach to leveraging experiences across users without sharing data would be to combine existing federated learning techniques with offline RL algorithms. *Is such a naïve federation strategy sufficient to learn an excellent federated policy collaboratively? Furthermore, is federation even necessary?* In this section, we aim to understand the challenges of federated offline RL with the goal of designing an algorithmic framework to address these challenges.

We start by illustrating the issues in designing a federated offline RL algorithm. We consider the Hopper environment from MuJoCo (Todorov et al., 2012), with $|\mathcal{N}| = 10$, $|\mathcal{D}_i| = 5000$, and we use the data from the D4RL dataset (Fu et al., 2020). However, instead of using the data generated by the same policy for all clients, we consider the setting where five clients use the data from the hopper-expert-v2 dataset (which was generated using a completely trained (expert) SAC policy) and five clients use the data from the hopper-medium-v2 dataset (which was generated using a partially trained (medium) policy achieving only a third of the expert performance). The clients and the server are unaware of the quality (expert or medium) of the data. Fig. 1 shows the performance comparison of multiple algorithms, where the mean and the standard deviation are calculated over 4 seeds.

**Combining All Data (Centralized):** Combining data and learning centrally is the ideal scenario in supervised learning. However, as seen in Fig. 1, performing centralized training over combined data generated using different behavior policies in offline RL can be detrimental. This is consistent with Yu et al. (2021) that proves that pooling data from behavior policies with different expertise levels can exacerbate the distributional shift between the learned policy and the individual datasets, leading to poor performance. Similar deterioration due to combining data has also been observed in other offline RL literature (Fujimoto & Gu, 2021; Kumar et al., 2020a).

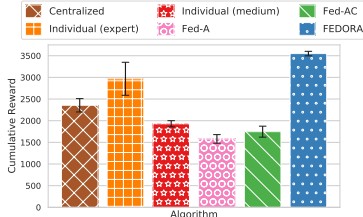

Figure 1: Performance comparison of federated and centralized offline RL algorithms.

**Individual Offline RL:** Here, agents apply offline RL to their own datasets without collaborating with others. In Fig. 1, we observe that clients with either expert or medium data do not learn well and exhibit a large standard deviation. This observation may be attributed to no client having sufficient data to learn a good policy.

**Naïve Federated Offline RL:** A simple federation approach is to use the offline RL objective as the local objective and apply FedAvg (Eq. (2)). However, offline RL algorithms typically comprise two components – an actor and a critic. It is unclear a priori which components should be federated, so we conduct experiments where we federate only the actor (Fed-A) or both the actor and the critic

(Fed-AC). Surprisingly, these naïve strategies result in federated policies that perform worse than individual offline RL, as witnessed in Fig. 1.

### 4.1 Issues with Federated Offline RL

Our example illustrates several fundamental issues that must be addressed while designing viable federated offline RL algorithms, including:

**1. Ensemble Heterogeneity:** Performing offline RL over heterogeneous data yields a set of policies of different qualities. It is crucial to leverage the information contained in these varied policies rather than simply averaging them. However, federation after a single-step local gradient at each client using weights in the manner of FedAvg, $w_i = |\mathcal{D}_i|/|\sum_{i=1}^{|\mathcal{N}|}|\mathcal{D}_i|$, is equivalent to solving the offline RL problem using the combined dataset of all clients (Wang et al., 2021). This approach leads to poor performance due to the resulting distribution shift, as shown in Fig. 1. *How should we optimally federate the ensemble of policies learned by the clients?*

**2. Pessimistic Value Computation:** Most offline RL algorithms involve a pessimistic term with respect to the offline data for minimizing the distribution shift. Training a client critic using only the local data with this pessimistic term could make it pessimistic towards actions poorly represented in its dataset but well represented in other clients' data. *How do we effectively utilize the federated critic along with the locally computed critic to set ambitious targets for offline RL at each client?*

**3. Data Heterogeneity:** Federated learning calls for performing multiple local gradient steps at each client before federation to enhance communication efficiency. However, numerous epochs would bias a client's local model to its dataset. This client drift effect is well known in federated (supervised) learning and could lead to policies that are not globally optimal. In turn, this could cause the federated policy's performance to be worse than training locally using only the client's data, as seen in Fig. 1. *How should we regularize local policies to prevent this?*

## 5 FEDORA Design Approach

We desire to develop a Federated Ensemble-Directed Offline RL Algorithm (FEDORA) that addresses the issues outlined in Section 4 in a systematic manner. Three fundamental requirements drive our approach. First, the clients jointly possess an ensemble of local policies of different (unknown) qualities, and the server must leverage the collective knowledge embedded in this ensemble during federation. Second, the quality of these policies must be assessed using an ensemble of critics that depend on local data for policy evaluation. Finally, after each round of federation, clients must update their local policies via offline RL utilizing both their local data and the received federated policy.

Maximizing the federated offline reinforcement learning (RL) objective in Eq. (4) using FedAvg would set weights as in Eq. (2), i.e., each client's contribution is weighted by the size of its dataset. This is is equivalent to solving the offline RL problem using the combined dataset of all clients. However, such an approach exacerbates the distribution shift problem that affects offline RL algorithms, leading to poor performance. This issue has been verified analytically and empirically in Yu et al. (2021). We illustrated this phenomenon in Fig. 1, where offline RL over pooled data resulted in a sub-optimal policy. The recommendation in Yu et al. (2021) is to share data conservatively by identifying which samples are likely to result in policy improvement. However, we cannot share any of the data across clients in the federated offline RL setting.

Our solution is to follow the principle of maximum entropy to choose weights that best represent the current knowledge about the relative merits of the clients' policies. Here, the weights are prevented from collapsing over a few clients that have the best current performance by adding an entropy regularization over the weights with temperature parameter $\beta$ resulting in the following objective:

$$U(\pi_{\text{fed}}) = \sum_{i=1}^{|\mathcal{N}|} w_i U_{\mathcal{D}_i}(\pi_{\text{fed}}) - \frac{1}{\beta} \sum_{i=1}^{|\mathcal{N}|} w_i \log w_i. \quad (5)$$

We can then show using a Lagrange dual approach that this objective is maximized when

$$w_i = \frac{e^{\beta U_{\mathcal{D}_i}(\pi_{\text{fed}})}}{\sum_{i=1}^{|\mathcal{N}|} e^{\beta U_{\mathcal{D}_i}(\pi_{\text{fed}})}}. \quad (6)$$

Based on these soft-max type of weights suggested by the entropy-regularized objective, we now design FEDORA accounting for each of the three requirements indicated above.

In what follows, $\pi_i^{(t,k)}$ denotes the policy of client $i$ in round $t$ of federation after $k$ local policy update steps. Since all clients initialize their local policies to the federated policy at the beginning of each round of federation, $\pi_i^{(t,0)} = \pi_{\text{fed}}^t$ for each client $i$. We also denote $\pi_i^t = \pi_i^{(t,K)}$, where $K$ is the maximum number of local updates. Since all clients initialize their local critics to the federated critic, we can similarly define $Q_i^{(t,k)}$, $Q_i^{(t,0)} = Q_{\text{fed}}^t$, and $Q_i^t = Q_i^{(t,K)}$ for the local critic.

## 5.1 Ensemble-Directed Learning over Client Policies

We first require a means of approximating $U_{\mathcal{D}_i}(\pi_{\text{fed}})$ in order to determine the weight $w_i$ of client $i$ as shown in Eq. (6). We utilize the performance of the final local policy $J_i^t = \mathbb{E}_{s \sim \mathcal{D}_i}\left[Q_i^t(s, \pi_i^t(s))\right]$, which also characterizes the relative performance at client $i$, as a proxy for $U_{\mathcal{D}_i}(\pi_{\text{fed}})$. Here, $Q_i^t$ is the local critic function at round $t$ after $K$ local updates. It is hard to directly obtain such an unbiased local critic $Q_i^t$ in offline RL, since we do not have access to the environment for executing the policy and evaluating its performance. Our approach toward computing $Q_i^t$ and $\pi_i^t$ are described later. The accuracy of the local estimates $J_i^t$ are highly dependent on the number of data samples available at $i$, and so in the usual manner of federated averaging, we need to account for the size of the dataset $|\mathcal{D}_i|$ while computing weights. We thus have client weights and federated policy update as

$$w_i^t = \frac{e^{\beta J_i^t}|\mathcal{D}_i|}{\sum_{i=1}^{|\mathcal{N}|} e^{\beta J_i^t}|\mathcal{D}_i|}, \quad \pi_{\text{fed}}^{t+1} = \sum_{i=1}^{|\mathcal{N}|} w_i^t \pi_i^t. \tag{7}$$

## 5.2 Federated Optimism for Critic Training

The critic in our algorithm plays two major roles. First, offline RL for policy updates at each client requires policy evaluation using local data. Second, policy evaluation by the critic determines weight $w_i^t$ of the local policy at client $i$ for ensemble learning during each round $t$ of federation. We desire a local critic at each client that can utilize the knowledge from the ensemble of critics across all clients while also being tuned to the local data used for policy evaluation.

A critic based on offline data suffers from extrapolation errors as state-action pairs not seen in the local dataset will be erroneously estimated, greatly impacting actor-critic style policy updates in federated offline RL. Since the federated policy is derived from the set of local policies, it may take actions not seen in any client's local dataset. This problem is exacerbated when the local policy at the beginning of each communication round is initialized to the federated policy. We introduce the notion of *federated optimism* to train local critics, wherein critics leverage the wisdom of the crowd and are encouraged to be optimistic. We achieve this federated optimism via two steps.

First, we use an ensemble-directed federation of the critics, where the local critic of client $i$ at round $t$ is weighed according to its merit to compute the federated critic as

$$Q_{\text{fed}}^{t+1} = \sum_{i=1}^{|\mathcal{N}|} w_i^t Q_i^t. \tag{8}$$

Such entropy-regularized averaging ensures that the critics from clients with good policies significantly influence the federated critic.

Second, for the local critic update, we choose the target value as the maximum value between the local critic and the federated critic, given by $\tilde{Q}_i^{(t,k)}(s, a) = \max\left(Q_i^{(t,k)}(s, a), Q_{\text{fed}}^t(s, a)\right)$, where $\tilde{Q}_i^{(t,k)}(s, a)$ is the target value of state $s$ and action $a$ at the $t^{\text{th}}$ round of federation after $k$ local critic updates. This ensures that the local critic has an optimistic (but likely feasible) target seen by the system. Using this optimistic target in the Bellman error, we update the local critic as

$$Q_i^{(t,k+1)} = \underset{Q}{\arg\min} \; \mathbb{E}_{(s,a,r,s') \sim \mathcal{D}_i}[(r + \gamma \tilde{Q}_i^{(t,k)}(s', a') - Q(s,a))^2], \tag{9}$$

where $a' = \pi_i^{(t,k)}$. In practice, we obtain $Q_i^{(t,k+1)}$ after a single gradient update.

## 5.3 Proximal Policy Update for Heterogeneous Data

While essential in order to set ambitious estimates, an optimistic critic might erroneously estimate the value of $\tilde{Q}_i^{(t,k)}$. Therefore, regularizing the local policy update w.r.t. both the local data and the federated policy is crucial. For regularization w.r.t. to the local offline data, we use the same method as in the TD3-BC algorithm and define the local loss function $\mathcal{L}_{\text{local}}(\pi) = \mathbb{E}_{(s,a) \sim \mathcal{D}_i}[-Q_i^{(t,k)}(s, \pi(s)) + (\pi(s) - a)^2]$. We then define the actor loss $\mathcal{L}_{\text{actor}}$ in Eq. (10), where the second term is a regularization w.r.t. to the federated policy. The local policy is updated using $\mathcal{L}_{\text{actor}}$,

$$\mathcal{L}_{\text{actor}}(\pi) = \mathcal{L}_{\text{local}}(\pi) + \mathbb{E}_{(s,a) \sim \mathcal{D}_i}[(\pi(s) - \pi_{\text{fed}}^t(s))^2], \quad \pi_i^{t,k+1} = \arg\min_\pi \mathcal{L}_{\text{actor}}(\pi). \quad (10)$$

## 5.4 Decaying the Influence of Local Data

FEDORA uses a combination of local data loss and a proximal term for its policy update Eq. (10). However, the local data loss might hamper the updated policy's performance since the local dataset may be generated according to a non-expert behavior policy. Hence, a client must decay the influence of its local data if it is reducing the performance of the updated policy by lowering the influence of $\mathcal{L}_{\text{local}}$ in $\mathcal{L}_{\text{actor}}$. To do so, we first evaluate the performance of the federated policy using the federated critic and local data at round $t$. For this evaluation, we use the proxy estimate $J_i^{\text{fed},t} = \mathbb{E}_{s \sim \mathcal{D}_i}[Q_{\text{fed}}^t(s, \pi_{\text{fed}}^t(s))]$. We compare this value with the performance of the updated policy, $J_i^t$, which is obtained using the updated critic. This difference provides us with an estimate of the improvement the local data provides. We decay the influence of $\mathcal{L}_{\text{local}}$ by a factor $\delta$ if $J_i^{\text{fed},t} \geq J_i^t$. We summarize FEDORA in Algorithm 1 and 2.

---

**Algorithm 1** Outline of Client $i$'s Algorithm

1: **function** train_client($\pi_{\text{fed}}^t, Q_{\text{fed}}^t$)
2:     $\pi_i^{(t,0)} = \pi_{\text{fed}}^t, \quad Q_i^{(t,0)} = Q_{\text{fed}}^t$
3:     **for** $1 \leq k < K$ **do**
4:         Update Critic by one gradient step w.r.t. Eq. (9)
5:         Update Actor by one gradient step w.r.t. Eq. (10)
6:     **end for**
7:     Decay $\mathcal{L}_{\text{local}}$ by $\delta$ if $J_i^{\text{fed},t} \geq J_i^t$
8: **end function**

---

**Algorithm 2** Outline of Server Algorithm

1: Initialize $\pi_{\text{fed}}^1, Q_{\text{fed}}^1$
2: **for** $t \in 1 \ldots$ **do**
3:     Send $\pi_{\text{fed}}^t$ and $Q_{\text{fed}}^t$ to $i \in \mathcal{N}$
4:     Sample $\mathcal{N}_t \subset \mathcal{N}$
5:     **for** $i \in \mathcal{N}_t$ **do**
6:         $i$.train_client ($\pi_{\text{fed}}^t, Q_{\text{fed}}^t$) (Client side)
7:     **end for**
8:     Compute $\pi_{\text{fed}}^{t+1}$ and $Q_{\text{fed}}^{t+1}$ for clients in $\mathcal{N}_t$ using Eq. (7) and (8) respectively.
9: **end for**

---

## 6 Experimental Evaluation

We conduct experiments to answer three broad questions: **(i) Comparative Performance:** How does FEDORA perform compared to other approaches with client data generated by heterogeneous behavior policies?, **(ii) Sensitivity to client updates and data quality:** How does the performance depend on the number of local gradient steps at clients, the randomness in the available number of agents for federation, and the quality of the data at the clients?, and **(iii) Ablation:** How does the performance depend on the different components of FEDORA? We implement FEDORA over the Flower federated learning platform (Beutel et al., 2020) which supports learning across devices. We also provide a simulation setup that can be executed on a single machine (See Appendix A).

**Baselines:** We consider the following baselines. **(i) Fed-A:** The local objective of all clients follows TD3-BC (Eq. 3). The server performs FedAvg over the actor's parameters, whereas each client learns the critic locally. **(ii) Fed-AC:** The local objective of all clients follows TD3-BC and the server performs FedAvg over the parameters of both the actor and the critic. **(iii) Fed-AC-Prox:** We add a proximal term to Fed-AC, which has been shown to help in federated supervised learning when clients have heterogeneous data (Li et al., 2020). **(iv) Heterogeneous Data-Aware Federated Learning (HDAFL)** We extend HDAFL (Yang et al., 2020) to the offline RL setting by dividing the actor network into generic and client-specific parts and then federating only the generic part during

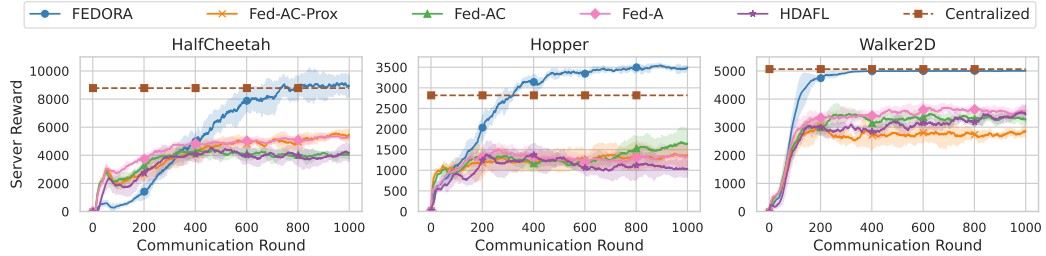

Figure 2: Evaluation of algorithms on different MuJoCo environments.

each round. **(v) Centralized:** We perform offline RL (TD3-BC) over the pooled data by combining the data present in all clients.

## 6.1 EXPERIMENTS ON SIMULATED ENVIRONMENTS

**Experimental Setup:** We focus on a scenario where clients are collaboratively learning to solve the same task, but the behavior policies used to collect data for each client could differ. We run experiments with the number of clients $|\mathcal{N}| = 50$, with each client having a local dataset of size $|\mathcal{D}_i| = 5000$. Of these 50 clients, 25 are provided with data from the D4RL (Fu et al., 2020) expert dataset, while the other 25 are provided with data from the D4RL medium dataset. The clients (and the server) are unaware of the quality of their datasets. Further, both the client and server do not have access to the environment. We choose $|\mathcal{N}_t| = 20$ clients at random to participate in each round $t$ of federation. The server obtains weights from clients in $|\mathcal{N}_t|$ and computes the federated weight $\pi_{\text{fed}}^{t+1}$ and $Q_{\text{fed}}^{t+1}$. For each plot, we evaluate the performance with four different seeds. We evaluate the performance of FEDORA and baselines over three MuJoCo tasks: Hopper, HalfCheetah, and Walker2D. During a round of federation, each client performs 20 epochs of local training in all algorithms, which is roughly 380 local gradient steps in our experimental setup.

**Comparative Performance of FEDORA:** In Fig. 2, we plot the cumulative episodic reward of the server/federated policy during each round of communication/federation. We observe that FEDORA outperforms all federated baselines and achieves performance equivalent to or better than centralized training. Furthermore, the federated baselines fail to learn a good server policy even after training for many communication rounds and plateau at lower levels compared to FEDORA, emphasizing that the presence of heterogeneous data hurts their performance.

To understand the effect of data coming from multiple behavior policies on centralized training, we consider a scenario where 50 clients with datasets of size $|D_i| = 5000$ participate in federation, with 25 clients having expert data and the other 25 having random data, i.e., data generated from a random policy. From Fig. 3, we notice that combining data of all clients deteriorates performance as compared to FEDORA. This observation highlights the fact that performing centralized training with data collected using multiple behavior policies can be detrimental.

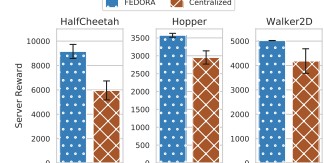

Figure 3: Comparison of FEDORA and centralized training with heterogeneous data.

**Sensitivity to Client Updates and Data Quality:** We study the sensitivity of FEDORA to client update frequency and data quality in the Hopper environment in the same setting as in Fig. 2. Increasing the number of local training steps can improve communication efficiency, but is detrimental under heterogeneous data due to client drift (Karimireddy et al., 2020). In Fig. 4(a), we study the effect of varying the number of local training epochs. We observe that increasing the number of epochs leads to faster learning, emphasizing that FEDORA can effectively learn with heterogeneous data. Not all clients may participate in every round of federation due to communication/compute constraints. In Fig.4(b), we study the effect of the fraction of clients participating in federation. We observe that FEDORA is robust towards variations in the fraction of clients during federation. Finally, in Fig. 4(c) we study the effect of data heterogeneity by varying the percentage of clients with expert datasets. We observe that FEDORA performs well even when only 20% of the total clients have expert-quality data. We present **several ablation studies and additional experiments** in appendix B.

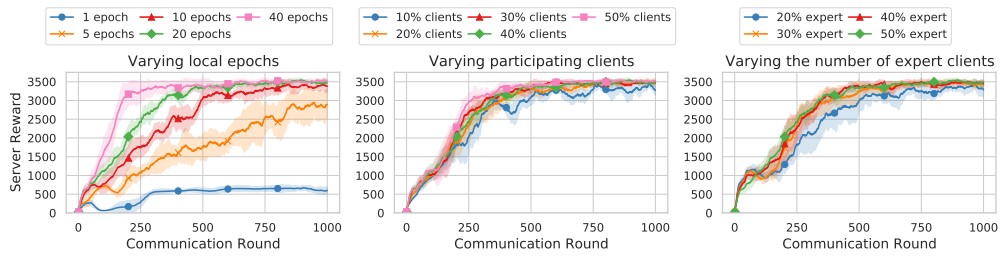

Figure 4: Effect of varying the number of (a) local gradient steps, (b) participating clients in each round, and (c) expert clients in FEDORA.

## 6.2 REAL-WORLD EXPERIMENTS ON TURTLEBOT

We evaluated the performance of FEDORA on TurtleBot (Amsters & Slaets, 2020), a two-wheeled differential drive robot (Fig. 5) to collaboratively learn a control policy to navigate waypoints while avoiding obstacles using offline data distributed across multiple robots (clients). This scenario is relevant to several real-world applications, such as cleaning robots in various houses, which aim to collaboratively learn a control policy to navigate and avoid obstacles using data distributed across different robots. Collaborative learning is essential, because a single robot might not have enough data to learn from or have encountered adequately different scenarios. Additionally, federated learning overcomes the privacy concerns associated with sharing data among the robots.

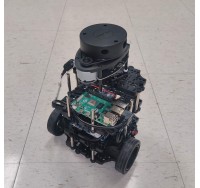

Figure 5: Turtle-Bot3 Burger.

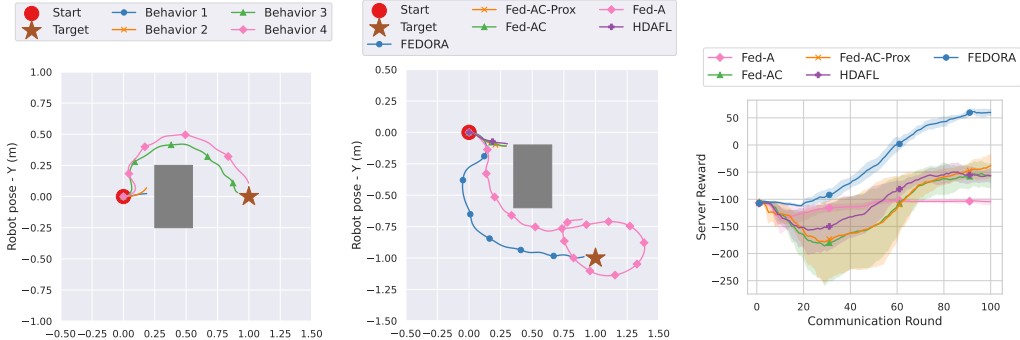

(a) Trajectories of behavior policies (b) Trajectories of learned policies (c) Comparison of FEDORA with federated baseline algorithms

Figure 6: Evaluation of FEDORA and other federated baselines for a mobile robot navigation task in the presence of an obstacle.

We collect data in the real-world using four behavior policies with varying levels of expertise ( Fig. 6(a)). We train over 20 clients for 100 communication rounds, each consisting of 20 local epochs (see Fig. 6(c)). Fig. 6(b) shows the trajectories obtained by the learned policies of different algorithms in the real-world, and only FEDORA is able to successfully reach the target by avoiding the obstacle. We provide more details in Appendix C. **A video of our experiment and code is provided in supplementary material.** We discuss limitations and societal impact of our work in Appendix D.

## 7 CONCLUSION

We presented an approach for federated offline RL, accounting for the heterogeneity in the quality of the ensemble of policies that generated the data at the clients. We solved multiple challenging issues by systematically developing a well-performing ensemble-directed approach entitled FEDORA, which extracts the collective wisdom of the policies and critics and discourages excessive reliance on irrelevant local data. We demonstrated its performance on several simulation and real-world tasks.

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

APPENDIX

We present several results and details in the appendix that illustrates the performance of FEDORA. These include details of our experimental setup (Appendix A), additional experiments studying different components of FEDORA and illustrating its performance in different settings (Appendix B), details of our real-world experiments using a TurtleBot (Appendix C), and discussion on limitations, societal impact and future work (Appendix D).

## A EXPERIMENTAL SETUP

**Algorithm Implementation:** We use the PyTorch framework to program the algorithms in this work, based on a publicly-available TD3-BC implementation. The actor and the critic networks have two hidden layers of size 256 with ReLu non-linearities. We use a discount factor of 0.99, and the clients update their networks using the Adam optimizer with a learning rate of $3 \times 10^{-4}$. For training FEDORA, we fixed the decay rate $\delta = 0.995$ and the temperature $\beta = 0.1$. TD3-BC trains for $5 \times 10^5$ time steps in the centralized setup. The batch size is 256 in both federated and centralized training.

The training data for clients are composed of trajectories sampled from the D4RL dataset. In situations where only a fraction of the clients partake in a round of federation, we uniformly sample the desired number of clients from the entire set.

**Federation Structure:** We implement FEDORA over the Flower federated learning platform (Beutel et al., 2020), which supports learning across devices with heterogeneous software stacks, compute capabilities, and network bandwidths. Flower manages all communication across clients and the server and permits us to implement the custom server-side and client-side algorithms of FEDORA easily. However, since Flower is aimed at supervised learning, it only transmits and receives a single model at each federation round, whereas we desire to federate both policies and critic models. We solve this limitation by simply appending both models together, packing and unpacking them at the server and client sides appropriately.

While 'FEDORA-over-Flower' is an effective solution for working across distributed compute resources, we also desire a simulation setup that can be executed on a single machine. This approach sequentially executes FEDORA at each selected client, followed by a federation step, thereby allowing us to evaluate the different elements of FEDORA in an idealized federation setup.

**Compute Resources:** Each run on the MuJoCo environments (as in Fig. 2) takes around 7 hours to complete when run on a single machine (AMD Ryzen Threadripper 3960X 24-Core Processor, 2x NVIDIA 2080Ti GPU). This time can be drastically reduced when run over distributed compute using the Flower framework.

## B ADDITIONAL EXPERIMENTS

### B.1 IMPORTANCE OF INDIVIDUAL ALGORITHM COMPONENT

We perform an ablation study to examine the different components of our algorithm and understand their relative impacts on the performance of the federated policy. We use the experimental framework with 10 clients and the Hopper environment described in Section 4, and plot the performance of the federated policy with mean and standard deviation over 4 seeds. The ablation is performed in two ways: (a) We build up FEDORA starting with Fed-A, the naïve method which federates only the actor, and add one new algorithm component at a time and evaluate its performance. (b) We exclude one component of FEDORA at a time and evaluate the resulting algorithm.

We observe in Fig. 7a that using priority-weighted averaging of the client's policy to compute the federated policy (Eq. (7)), and an optimistic critic (Eq. (8) - (9)) significantly improves the performance of the federated policy. This is consistent with our intuition that the most important aspect is extracting the collective wisdom of the policies and critics available for federation, and ensuring that the critic sets optimistic targets. The proximal term helps regularize local policy updates (Eq. (10)) by choosing actions close to those seen in the local dataset or by the federated policy.

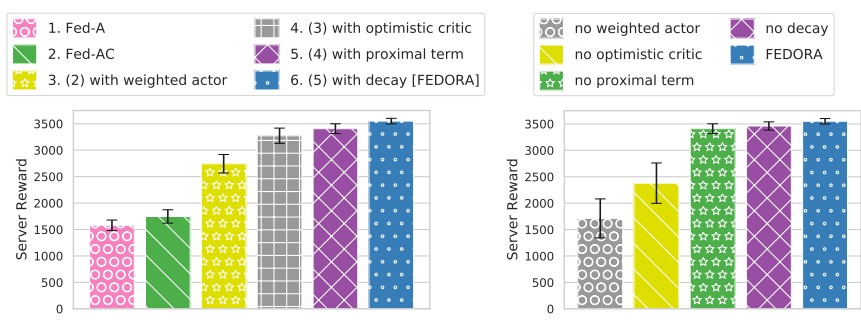

(a) Effect of sequentially adding one algorithm component at a time

(b) Effect of removing one individual algorithm components from FEDORA

Figure 7: Ablation Studies.

Additionally, decaying the influence of local updates enables the local policy to leverage the federated policy's vantage by choosing actions not seen in the local dataset.

From Fig. 7b, we observe that removing priority-weighted actor from FEDORA causes the steepest drop in performance, followed by the optimistic critic. Again, this is consistent with our intuition on these being the most important effects. Excluding the proximal term and local decay also results in a reduction in server performance along with a greater standard deviation.

### B.1.1 ABLATION OF DECAYING MECHANISM ON WALKER ENVIRONMENT

We study the effect of decaying the influence of local data (5.4) in the Walker2D environment in Figure 8. Although the decaying mechanism seems to give only a small improvement in Figure , which pertains to experiments on the Hopper environment, we observe that it provides a significant improvement in the Walker2D environment.

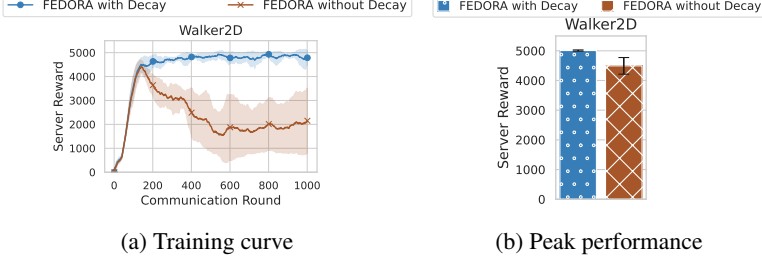

(a) Training curve

(b) Peak performance

Figure 8: Ablation study of decaying mechanism on Walker2d environment (setting similar to Fig 7).

### B.2 ANALYSIS OF CLIENT PERFORMANCE

We train FEDORA on MuJoCo environments using a setup similar to Section 6 where 20 out of the 50 clients are randomly chosen to participate in each round of federation. Our goal is to analyze the contribution of clients with expert data and those with medium data to the learning process. As before, the clients and the algorithm are unaware of the data quality.

We plot the mean weights $w_i^t$ across the expert and medium dataset clients participating in a given round of federation in Fig. 9a. We observe that the weights of medium clients drop to $0$, while the weights of expert clients rise to $0.1$. This finding emphasizes the fact that clients are combined based on their relative merits.

In Fig. 9b, we plot the mean of the decay value associated with $\mathcal{L}_{\text{local}}$ across participating expert and medium dataset clients (Section 5.4). The decay of both sets of clients drops as training progresses. A reduction in decay occurs each time the local estimate of the federated policy's performance $J_i^{\text{fed},t}$ is greater than the estimated performance of the updated local policy $J_i^t$. A decreasing decay implies

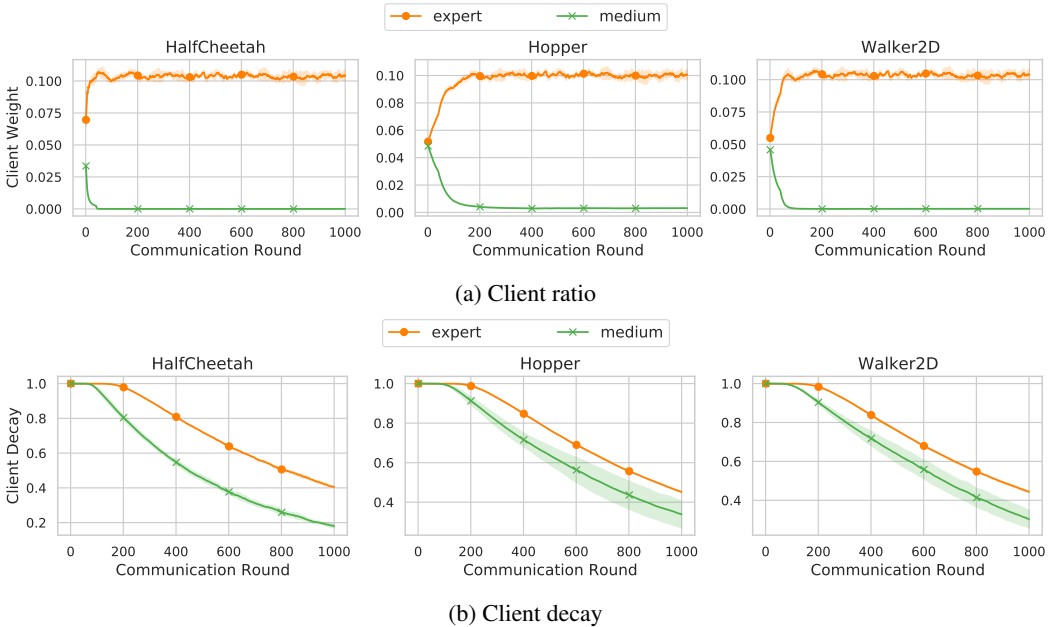

(a) Client ratio

(b) Client decay

Figure 9: Analysis of client performance during federation. The average of the performance metric is computed across expert and medium clients participating in a given round of federation.

that the federated policy offers a performance improvement over local policies more often as the rounds $t$ advance. Thus, training only on local data is detrimental, and participation in federation can help learn a superior policy.

## B.3 FEDERATED OFFLINE RL EXPERIMENTS WITH CITYLEARN

Real-world environments often have a large state space and are stochastic in nature. We run federated experiments on CityLearn (Vázquez-Canteli et al., 2020) to assess the effectiveness of FEDORA on such large-scale systems. CityLearn is an OpenAI Gym environment with the goal of urban-scale energy management and demand response, modeled on data from residential buildings. The goal is to reshape the aggregate energy demand curve by regulating chilled water tanks and domestic hot water, two modes of thermal energy storage in each building. The energy demand of residential buildings changes as communities evolve and the weather varies. Hence, the controller must update its policy periodically to perform efficient energy management. Federated learning would allow utilities that serve communities in close proximity to train a policy collaboratively while preserving user data privacy, motivating the use of FEDORA for this environment.

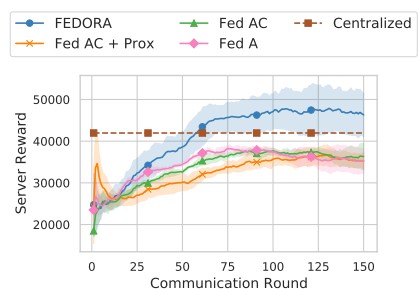

Figure 10: Evaluation of algorithms on CityLearn.

In our experiments, we have 10 clients with 5000 training examples such that they all participate in 150 rounds of federation. The training data for the clients is obtained from NeoRL, an offline RL benchmark Qin et al. (2021). 5 clients each have data from the CityLearn High and CityLearn Low datasets, which are collected by a SAC policy trained to 75% and 25% of the best performance level, respectively. During each round of federation, each client performs 20 local epochs of training. The server reward at the end of each federation round is evaluated online and shown in Fig. 10. We observe that FEDORA outperforms other federated offline RL algorithms as well as centralized training, which learns using TD3-BC on the data aggregated from every client. These findings indicate that FEDORA can perform well in large-scale stochastic environments.

### B.4 Effect of multiple behavior policies and proportion of clients participating in federation

In this section, we study the effects of clients having data from multiple behavior policies for varying proportions of clients participating in federation. We consider a scenario with 50 clients having $D_i = 5000$ in the Hopper-v2 environment where,

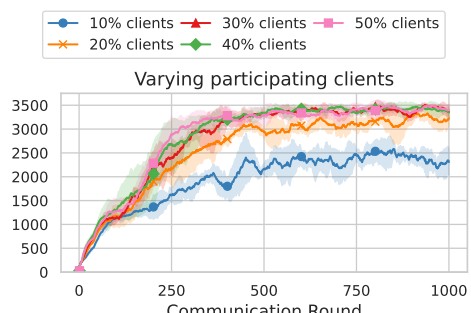

- 12 clients have expert data (samples from a policy trained to completion with SAC.).
- 12 clients have medium data (samples from a policy trained to approximately 1/3 the performance of the expert).
- 14 clients have random data ( samples from a randomly initialized policy).
- 12 clients have data from the replay buffer of a policy trained up to the performance of the medium agent.

Figure 11: Effect of varying the number of participating clients in each round on FE-DORA

We run FEDORA by varying the the percentage of clients participating in each round of federation. We observe that the FEDORA is fairly robust to the fraction of clients participating in federation even when the fraction is as low as 20%.

### B.5 Centralized training with other Offline RL algorithms

We consider a scenario similar to the one in Fig. 3 for the Hopper-v2 environment with 50 clients, having $|D_i| = 5000$ participating in federation, where 25 clients have expert data, and 25 clients have random data. We compare the performance of different Offline RL algorithms over the pooled data with FEDORA. The algorithms we choose are Conservative Q-Learning for Offline Reinforcement Learning (CQL) Kumar et al. (2020b) and Offline Reinforcement Learning with Implicit Q-Learning (IQL) Kostrikov et al. (2021) whose implementations are obtained from the CORL library Tarasov et al. (2022). We can observe from Fig. 12 that pooling data from different behavior policies affects both offline RL algorithms.

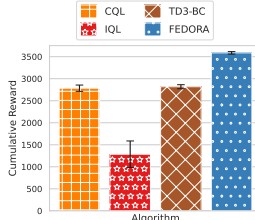

Figure 12: Comparison with different Offline RL algorithms

## C Details of Real-World Robot Experiments

### C.1 Demonstration Data Collection

We train four behavior policies of varying levels of expertise using TRPO Schulman et al. (2015) on a custom simulator for mobile robots described in section C.2. The first policy is capable of waypoint navigation but collides with obstacles. The second policy can reach waypoints while avoiding obstacles present at one fixed position. The third policy has not fully generalized to avoiding obstacles at various positions. Finally, the fourth policy can navigate to the goal without any collision. We execute the behavior policies in the real-world by varying the waypoint (target location) and location of the obstacle to gather demonstration data, which we then use to train FEDORA and other baselines. Each client has a dataset consisting of 300 data points collected using a single behavior policy. After training, we test the learned policies in the real-world on a TurtleBot to ascertain its feasibility.

### C.2 Simulator Design

We develop a first-order simulator for mobile robots using the OpenAI Gym framework, which enables the training of RL algorithms. The robot's pose is represented by its X- and Y-coordinates in

a 2D space and its orientation with respect to the X-axis, $\theta$. The pose is updated using differential drive kinematics

$$
\begin{array}{rcl}
x_{t+1} & = & x_t + \Delta t\, v \cos \theta_t \\
y_{t+1} & = & y_t + \Delta t\, v \sin \theta_t \\
\theta_{t+1} & = & \theta_t + \Delta t\, \omega,
\end{array}
\tag{11}
$$

where $(x_t, y_t, \theta_t)$ is the pose at time $t$, $v$ and $w$ are the linear and angular velocity of the robot respectively, and $\Delta t$ is time discretization of the system.

The simulator uses a functional LIDAR to detect the presence of obstacles. We simulate the LIDAR using a discrete representation of the robot and obstacles in its immediate environment. For each scanning direction around the LIDAR, we use Bresenham's line algorithm to generate a path comprising of discrete points. The simulator determines LIDAR measurements by counting the number of points along each path, starting from the robot and continuing until it encounters and obstacle or reaches the maximum range.

The reward function is designed to encourage effective waypoint navigation while preventing collisions. We define a boundary grid that extends for 1m beyond the start and the goal positions in all directions. The reward function at time $t$ for navigating to the goal position $(x_g, y_g)$ is chosen to be

$$
R_t = \begin{cases}
+100, & \text{if } |x_t - x_g| \le thresh \text{ and } |y_t - y_g| \le thresh \\
-10, & \text{if robot outside boundary} \\
-100, & \text{if robot collides} \\
-(\texttt{c.t.e}_t^2 + \texttt{a.t.e}_t + \texttt{h.e}_t) + \sum \texttt{lidar}_t, & \text{otherwise}
\end{cases}
\tag{12}
$$

where $\texttt{c.t.e}_t$ is the cross-track error, $\texttt{a.t.e}_t$ is the along-track error, $\texttt{h.e}_t$ is the heading error, $\texttt{lidar}_t$ is the array of LIDAR measurements at time $t$, and $thresh$ is the threshold error in distance, chosen as 0.1m. Let the L-2 distance to the goal and the heading to the goal at time $t$ be $d_t^g$ and $\theta_t^g$ respectively. Then, we have

$$
\begin{array}{rcl}
d_t^g & = & \sqrt{(x_g - x_t)^2 + (y_g - y_t)^2}, \\
\theta_t^g & = & \tan^{-1}\left(\frac{y_g - y_t}{x_g - x_t}\right), \\
\texttt{c.t.e}_t & = & d_t^g \sin(\theta_g - \theta_t), \\
\texttt{a.t.e}_t & = & |x_g - x_t| + |y_g - y_t|, \\
\texttt{h.e}_t & = & \theta_t^g - \theta_t.
\end{array}
\tag{13}
$$

### C.3 MOBILE ROBOT PLATFORM

We evaluate the trained algorithms on a Robotis TurtleBot3 Burger mobile robot (Amsters & Slaets, 2020), an open-source differential drive robot. The robot has a wheel encoder-based pose estimation system and is equipped with an RPLIDAR-A1 LIDAR for obstacle detection. We use ROS as the middleware to set up communication. The robot transmits its state (pose and LIDAR information) over a wireless network to a computer, which then transmits back the corresponding action suggested by the policy being executed.

## D LIMITATIONS, SOCIETAL IMPACTS, AND FUTURE WORK

### D.1 LIMITATIONS AND FUTURE WORK

In this work, we examine the issue of Federated Offline RL. We make the assumption that all clients share the same MDP model (transition kernel and reward model), and any statistical variances between the offline datasets are due to differences in the behavior policies used to collect the data. Moving forward, we aim to broaden this to cover scenarios where clients have different transition and reward models. To achieve this, we plan to extend ideas from offline meta RL to the federated learning scenario. Furthermore, we plan to explore personalization in federated offline RL as an extension to our research. We also believe that our approach may also be useful in the context of federated supervised learning, especially when the data is sourced from varying qualities, and we intend to formally investigate this in the future as a seperate line of work.

### D.2 Ethics Statement and Societal Impacts

In this work, we introduce a novel algorithm for federated offline reinforcement learning. The domain of federated offline RL offers the potential for widespread implementation of RL algorithms while upholding privacy by not sharing data, as well as reducing the need for communication. Throughout our study, no human subjects or human-generated data were involved. As a result, we do not perceive any ethical concerns associated with our research methodology.

While reinforcement learning holds great promise for the application in socially beneficial systems, caution must be exercised when applying it to environments involving human interaction. This caution arises from the fact that guarantees in such scenarios are probabilistic, and it is essential to ensure that the associated risks remain within acceptable limits to ensure safe deployments.

## E ICLR 2024 Rebuttal

### E.1 Data Rebalancing

We consider a scenario similar to Fig. 1, where we consider the Hopper-v2 environment with 10 clients having $|D_i| = 5000$ participating in federation, where 5 clients have expert data, and 5 clients have medium data. We compare the performance of TD3-BC, TD3-BC with data rebalancing (TD3-BC_RB) (Yue et al., 2022; 2023), and FEDORA. From Fig. 13 we can notice that the addition of data rebalancing does help the performance of offline RL algorithms when data is collected using multiple behavior policies. We also notice that the performance of TD3-BC with data rebalancing does not match the performance of FEDORA. We hypothesise that this could be due to the superior weighting mechanism employed by FEDORA, and that data rebalancing cannot completely solve the distribution shift issue caused by data coming from multiple behavior policies.

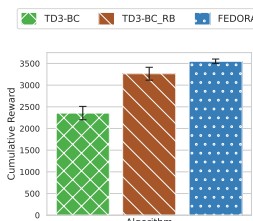

Figure 13: Comparison with Offline RL with data rebalancing

### E.2 Different Weighing Mechanisms

We consider a scenario similar to Fig. 1, where we consider the Hopper-v2 environment with 10 clients having $|D_i| = 5000$ participating in federation, where 5 clients have expert data, and 5 clients have medium data. We conduct two experiments, **(1.) RC:** We combine clients based on the average reward in their dataset using the weighing scheme proposed in (Yue et al., 2022). In this scenario, we choose the weights of federation for client $i$, $w_i = \frac{p_i}{\sum_{k \in \mathcal{N}_t} p_k}$. Where $p_i = \frac{R_i - R_{\min}}{R_{\max} - R_{\min}}$, and $\mathcal{N}_t$ is the set of clients participating in federation at round $t$. Here $R_i$ corresponds to the average reward of client $i$'s dataset, $R_{\min} = \min_{i \in \mathcal{N}_t} R_i$, and $R_{\max} = \max_{i \in \mathcal{N}_t} R_i$. Each client runs TD3-BC as their local offline algorithms and does not employ any of the strategies discussed in section 5. **(2.) FEDORA_RC:** We modify the weighting strategy of FEDORA to the one described above. This method uses the weighing strategy described in RC. We keep all the other features of FEDORA.

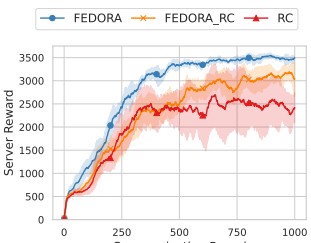

Figure 14: Comparison with different a weighing mechanism

From Fig. 14 we notice that FEDORA outperforms both baselines, and the addition of the different components of FEDORA improves the performance of RC. FEDORA outperforms the baselines due to its superior weighing strategy, and the holistic approach we took to designing it by augmenting different algorithmic components that help mitigate the problems of federated offline RL described in section 4.1.

### E.3 $\beta$ Hypterparameter sweep

In Fig. 15 run FEDORA for different values of $\beta$, which is the temperature parameter of federation. We consider a scenario similar to Fig. 1, where we consider the Hopper-v2 environment with 10 clients having $|D_i| = 5000$ participating in federation, where 5 clients have expert data, and 5 clients have medium data. When $\beta = 0$, it boils down to a uniform weighting scheme, where the quality of data present in each client is not considered during federation. As $\beta \to \infty$ it tends to a max weighting scheme, where the federated policy is the same as an individual client's policy with the highest quality data.

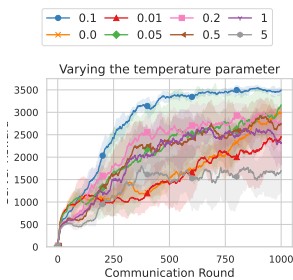

Figure 15: Varying $\beta$

