# OpenReview forum: "Federated Ensemble-Directed Offline Reinforcement Learning"
_ICLR.cc/2024/Conference — Submitted to ICLR 2024_

### Official Review · Reviewer_mieC · 2023-10-30

**Soundness:** 3 good
**Presentation:** 3 good
**Contribution:** 3 good
**Rating:** 6
**Confidence:** 3

**Summary:**

The paper considers federated reinforcement learning problems with offline datasets of different qualities.
Instead of aggregating models in a uniform way, FEDORA adopts the principle of maximum entropy to aggregate local models accordingly. Specifically, the model from offline datasets with higher quality weights more in the central aggregation.
It is empirically shown that FEDORA outperforms FedAvg in both simulated environments and real-world experiments.

**Strengths:**

1. The design of FEDORA is well-written and easy to follow.
2. The paper considers a novel offline federated reinforcement learning where local datasets are generated by different behavioral policies.
3. FEDORA has been evaluated in various environments and has achieved outstanding performance w.r.t. traditional FedAvg.

**Weaknesses:**

1. What role does the entropy regularizer in Eq. (5) play? Collapsing to datasets with the best performance does not seem to worsen the convergent performance of FEDORA. Is there any ablation study on this term?
2. The approach toward computing $Q_{t_i}$ and $\pi_{t_i}$ in Section 5.1 is not clearly described. The evaluation of Q functions and policy may be inaccurate on datasets of low quality. How will it influence the performance of FEDORA?

**Questions:**

See Weaknesses.

---

> ### Author Response · Authors · 2023-11-21
> **Response to reviewer mieC**
>
> Thank you very much for your comments and suggestions. We are delighted to know that you find our paper well written, our framework novel, and our experiments extensive.
>
>  In response to your question regarding the entropy regularizer paramater $\beta$, **we have now conducted experiments**, please see **Appendix E.3**.  We believe that we have addressed all your concerns, and we sincerely hope that the reviewer would consider increasing their score.
>
> Below we the concerns raised in your review.
>
> **Q1.** *``What role does the entropy regularizer in Eq. (5) play''*
>
> **Response:** The role of the entropy regularizer is to modify the ensemble scheme of our algorithm.  When $\beta=0$, it reduces to a uniform weighting scheme, where the quality of data present in each client is not considered during federation. As $\beta \rightarrow \infty$ it tends to a max weighting scheme, where the federated policy is the same as an individual client's policy with the highest quality data. In response to your question, we have **now conducted additional experiments** with a hyperparameter sweep for different values of $\beta$, see Appendix E.3.
>
>
> **Q2.** *``The approach toward computing $Q_t$ and $\pi_t$ in Section 5.1 is not clearly described''*
>
> **Response:** $Q_i^t$ and $\pi_i^t$ are the local critic and actor at client $i$ in round $t$. It is computed by minimizing the loss in Eq. 9 and Eq. 10 respectively. We follow an ensemble learning approach where in we weigh clients according to the quality of their datasets, and initialize the local critic and actor during each round of federation to the federated actor and critic. This ensures that each client starts off with the knowledge possessed by clients participating in federation. We then eventually down weigh clients who perform poorly  and decay the influence of local data. Thus, although a client individually has low quality data, using actor and critic trained over the ensemble to initialize the training process, and decaying the influence of local data, ensures the clients of datasets of low quality do not affect the performance of FEDORA. Please note that we have already explained these steps modularly in Section 5.1, 5.2, 5.3 and 5.4.

---

### Official Review · Reviewer_oNBT · 2023-10-31

**Soundness:** 2 fair
**Presentation:** 2 fair
**Contribution:** 2 fair
**Rating:** 3
**Confidence:** 4

**Summary:**

This paper addresses the problem of federated offline reinforcement learning, aiming to learn a high-performing policy by leveraging multiple distributed offline datasets without the need for centralizing the data. The authors highlight several challenges inherent in this setup and propose a framework called FEDORA (Federated Ensemble-Directed Offline Reinforcement Learning Algorithm) to tackle these challenges in the context of federated offline reinforcement learning. To evaluate the effectiveness of FEDORA, the authors conduct experiments on mujoco environments, demonstrating its capability to achieve desirable results. Furthermore, the proposed framework is deployed on real turtlebots, providing practical validation of its performance and applicability in real-world scenarios.

**Strengths:**

This paper addresses the intriguing problem of federated offline reinforcement learning, which has received limited attention in the existing literature. The authors shed light on various technical challenges that emerge from this unique setting. To tackle these challenges, they propose an innovative algorithm called FEDORA. The efficacy of FEDORA is demonstrated through its successful application to continuous control tasks in mujoco environments, showcasing its ability to learn effective policies. Moreover, the authors validate the practicality and real-world applicability of the proposed framework by deploying it on real turtlebots, further highlighting its performance under realistic scenarios.

**Weaknesses:**

While the paper introduces a novel approach to an interesting problem, there are several areas that could benefit from further improvement:
1. The assumption of identical MDPs and reward functions across all agents limits the generalizability and applicability of the proposed approach to real-world scenarios. Exploring techniques to handle heterogeneity among agents' environments could enhance the practicality of the framework.
2. The requirement for the server to have access to the complete MDP raises concerns about privacy and may not be practical in many scenarios. Providing a practical example or discussing potential alternatives for this setup would strengthen the paper's applicability.
3. The paper assumes **knowledge of dataset quality**, but it does not thoroughly address the issue of comparing and assessing the quality of different datasets. Specifically, in Sec. 4 , is the dataset generated by hopper-expert-v2 guaranteed to be better than that generated by hopper-medium-v2? Suppose that $D_m$ is generated by the medium policy which has converged while $D_e$ is generated by the expert policy during the initial random exploration stage. Is the quality of $D_e$ always better than $D_m$ ? It is important to explore methods to evaluate dataset quality more robustly and consider scenarios where the assumption of one dataset being consistently better than another may not hold.
4. Experimental details for Figure 1 are missing, making it difficult to fully understand and interpret the results. Including the specific experimental setup, including hyperparameters, training procedures, and any other relevant details, would enhance the reproducibility and credibility of the findings.
5. Eq. (7) implies that the server selects the action based on a weighted average of all agents' actions. Further clarification is needed to explain how the weights assigned to individual agents' actions influence the final decision. Does the weighting scheme prevent one agent from significantly overpowering the others? If not, what mechanisms or strategies are in place to address potential dominance issues and ensure a balanced contribution from all participating agents?
6. The process of training the centralized policy to achieve the straight-line performance in Figure 2 requires further explanation.
7. Can you compare the performance of FEDORA with just the expert policy running the offline RL algorithm in Fig. 2?
8. Some important FedRL papers are missing from reference. Including them would strengthen the scholarly contribution of the paper.

**Questions:**

1. Could you provide a compelling real-world example that demonstrates the practicality of the problem setup? By illustrating a specific scenario or application where the proposed approach can be effectively applied, readers can gain a better understanding of its relevance and potential impact.
2. How do you determine the quality of a given dataset?
3. How is the value of $\beta$ determined, and what insights can you provide regarding its impact on the weights assigned to individual agents?
4. Could you provide a brief discussion or analysis of the computational and communication costs associated with the proposed approach?

---

> ### Author Response · Authors · 2023-11-21
> **Response to Reviewer oNBT (1/2)**
>
> This is a **copy-paste review** of our earlier NeurIPS submission.  We are afraid that the review for our NeurIPS submission was full of misconceptions, which are repeated in this ICLR review.  The reviewer had not responded to our clarifications for NeurIPS, and we sincerely hope that the reviewer will find time to respond to our clarifications this time around, and reconsider and increase their score.  Thank you for your time.
>
> **Q1.** *``The requirement for the server to have access to the complete MDP raises concerns...''*
>
> **Response:** **We are afraid that the reviewer had a misunderstanding here.** We **do not assume** that the server has access to the MDP, simulator or any data present in the clients. Furthermore, the clients do not have access to the MDP or a simulator.  Indeed, this is the fundamental premise of offline RL. We would appreciate it if you could point out the lines in our paper that caused the misunderstanding, and we will be happy to fix them.
>
> **Q2.** *``The paper assumes knowledge of dataset quality, but ..''*
>
> **Response:**  **We are afraid that the reviewer had a misunderstanding here.** In our setting, the clients or server are **not aware** of the quality of the data they possess. Indeed, the lack of this knowledge is why we develop the ensemble-directed learning approach, under which estimating the quality of local polices and critics is critical, and which is a fundamental contribution of our work.  We would appreciate it if you could point out the lines in our paper that caused the misunderstanding, and we will be happy to fix them.
>
> **Q3.**  *``Experimental details for Figure 1 are missing, making it difficult to ...''*
>
> **Response:** The basic description of the experimental setting for the results shown in Figure 1 is provided in Section 4 right above Fig. 1. More detailed description of all our experiments are given in Section 6 and in Appendix A. We have also provided the entire code base from where  you can run the experiments in the supplementary material.
>
> **Q4.** *``Eq. (7) implies that the server selects the action based on a weighted average of all agents' actions.''*
>
> **Response:**  **We are afraid that the reviewer had a misunderstanding here.**  **The server does not take any action at all.**  As with all federated learning approaches, the server merely computes the federated policy using the local policies of the clients, which  is then sent back to the clients (see algorithms 1 and 2). The weights are based on the estimated quality of the data present in each client, and the weighting scheme ensures that the well performing policies are given more importance.
>
> **Q5.** *``The process of training the centralized policy to achieve the straight-line performance in Figure 2 requires further explanation.''*
>
> **Response:** Please note that the straight line is not a training curve, rather the performance of the policy trained using TD3-BC on the combined data present in all clients.  This is mentioned in Section 6.
>
> **Q6.** *``Some important FedRL papers are missing from reference. Including them would strengthen the scholarly contribution of the paper.''*
>
> **Response:** We did a through literature survey and cited the most of relevant papers that we are aware of. We will be happy to include more papers in our literature survey if the reviewer could kindly give pointers to any relevant papers that are missing from our references.
>
> **Q7.**  *``Can you compare the performance of FEDORA with just the expert.''*
>
> **Response:** Please note that we have indeed done this.  In  Figure 1,  we compare the performance of FEDORA with the performance of the policy obtained by training on the expert data present in an individual client.
>
>  **Q8.**  *``Could you provide a compelling real-world example that demonstrates the practicality $\dots$*
>
> **Response:** We illustrated the value proposition of the approach using a mobile robotics application, wherein each robot collects data independently and privately via arbitrary policies, and our approach is successful at distilling their collective knowledge of navigation without sharing any data (video in supplementary material).  The scenario has several applications, such as that of cleaning robots, which must collaboratively learn to navigate within homes, without sharing data that might violate home owners' privacy.  Much the same scenario applies to factory floor or warehouse robots that must collaboratively learn to navigate, without sharing data that would stress the limited network bandwidth available.
>
> **Q9.** *``How do you determine the quality of a given dataset?''*
>
> **Response:** We use a variant of off-policy evaluation, where the clients uses its local data and the federated/local critic to evaluate the quality of its dataset. (See section 5.1 and 5.2)

---

> ### Author Response · Authors · 2023-11-21
> **Response to Reviewer oNBT (2/2)**
>
> **Q10.** *``How is the value of $\beta$ determined''*
>
> **Response:** In our setup, we treat $\beta$ as a hyperparameter and use a fixed value of $0.1$ for all our experiments. When $\beta=0$, it boils down to a uniform weighting scheme, where the quality of data present in each client is not considered during federation. As $\beta \rightarrow \infty$ it tends to a max weighting scheme, where the federated policy is the same as an individual client's policy.
>
>
> **Q11.**  *``Could you provide a brief discussion  or analysis of the computational and communication $\dots$"*
>
> **Response:**  The computation and communication costs are similar to standard federated learning algorithms. That is why we omitted the details. In short:
>
> **Client side:** A subset of participating clients initializes its local policy to the federated policy and performs gradient updates using local data, and evaluates the performance of the learnt policy using the local data. **Server side:** Performs a weighted-averaging of the weights of the actor and critics of the participating clients. **Communication:** The subset of participating clients communicates back the weights of actor and critic to the server. The server broadcasts the federated actor and critic to all clients.

---

### Official Review · Reviewer_RUBc · 2023-10-31

**Soundness:** 3 good
**Presentation:** 3 good
**Contribution:** 3 good
**Rating:** 6
**Confidence:** 4

**Summary:**

The paper proposes a novel federated ensemble-directed offline reinforcement learning algorithm (FEDORA) that enables multiple agents with heterogeneous offline datasets to collaboratively learn a high-quality control policy without sharing data. The key contributions are:
1. Identifying key challenges of federated offline RL including ensemble heterogeneity, pessimistic value computation, and data heterogeneity.
2. Systematically addressing these challenges through an ensemble-directed federation approach that extracts collective wisdom of policies and critics and discourages over-reliance on possibly irrelevant local data.
3. Demonstrating strong empirical performance of FEDORA over baselines on MuJoCo environments and a real-world robot navigation task.

**Strengths:**

1. The paper clearly motivates the problem of federated offline RL and identifies key challenges that are not addressed by naively combining existing methods.
2. FEDORA is systematically designed to address the identified challenges in an intuitive manner through techniques like ensemble-directed federation and federated optimism.
3. Extensive experiments demonstrate the effectiveness of FEDORA over baselines on simulated and real-world tasks. The ablation studies provide useful insights.
4. The paper is clearly written and provides sufficient details to understand the proposed techniques.

**Weaknesses:**

1. How does FEDORA perform when some clients have near-random or adversarial datasets?
2. Have the authors experimented with different values of β and δ? Is there a principle behind setting them?
3. How does FEDORA compare to state-of-the-art offline RL algorithms like CQL or IQL in the federated setting?

The paper makes solid contributions in identifying and addressing key challenges in federated offline RL. The algorithm design is methodical and supported through extensive experiments. More theoretical and implementation details will further improve the paper.

**Questions:**

see above。

---

> ### Author Response · Authors · 2023-11-21
> **Response to Reviewer RUBc**
>
> We thank the reviewer for their valuable feedback. We are encouraged to hear that our *"paper makes solid contributions in identifying and addressing key challenges in federated offline RL"* and our *"algorithm design is methodical and supported through extensive experiments."*
>
> In response to your question regarding the hyper-parameter $\beta$ we have **conducted additional experiments** by varying it in the Hopper-v2 environment, please see Appendix E.3.
>
> Below, we give the detailed response to your comments. We believe that we have addressed all your concerns, and we sincerely hope that the reviewer would consider increasing their score.
>
> **Q1.**  *``How does FEDORA perform when some clients have near-random or adversarial datasets?''*
>
> **Response:** Thank you for your question. In our paper, we have already shown the the performance of FEDORA when some clients have random data. Specifically,   Fig.3 shows the superior performance of FEDORA when $25$ clients have random data and $25$ clients have expert data. Further, Fig.11  in the appendix shows the  effects of clients having data from multiple behavior policies for varying proportions of clients participating in federation. Here, 12 clients have expert data, 12 clients have medium data, 14 clients have random data, and 12 clients have data from the replay buffer of a policy trained up to the performance of the medium agent. We observe that FEDORA performs well even this scenario with clients having data from multiple policies.
>
>
> **Q2.**  *``Have the authors experimented with different values of $\beta$ and $\delta$? Is there a principle behind setting them?''*
>
> **Response:** Thank you for your question. In our setup, we treat $\beta$ as a hyperparameter and use a fixed value of $0.1$ for all our experiments. In response to your question, we have now conducted additional experiments with different values of $\beta$, see  Appendix E.3.  When $\beta=0$, it reduces to a uniform weighting scheme, where the quality of data present in each client is not considered during federation. As $\beta \rightarrow \infty$ it tends to a max weighting scheme, where the federated policy is the same as an individual client's policy with maximum value. We would like to point out that we do not have a $\delta$ parameter in our work.
>
>
> **Q3.** *``How does FEDORA compare to state-of-the-art offline RL algorithms like CQL or IQL in the federated setting?''*
>
> **Response:**  We designed FEDORA as a framework where we can use use any actor-critic based offline RL algorithm in the clients. We chose TD3-BC due to its simplicity and superior performance over other baselines such as CQL, AWAC, Fisher-BRC, and BRAC-Q. We kept the local offline RL algorithm consistent throughout our simulations for fair comparisons. We believe that if we use a superior local offline RL algorithm, the performance of FEDORA might be better.

---

### Official Review · Reviewer_MJbF · 2023-10-31

**Soundness:** 3 good
**Presentation:** 3 good
**Contribution:** 3 good
**Rating:** 5
**Confidence:** 3

**Summary:**

This study examines the issue of training offline RL in a federated environment. Initially, the authors outline the difficulties associated with federated offline RL. Following this, a method named FEDORA is introduced, which aims to enhance the collective performance of federated learning. FEDORA accomplishes this by managing how models are aggregated and by reducing the influence of local training data on the global model.

**Strengths:**

- Clarity and comprehensibility: The manuscript is articulated clearly. The content is easy to understand.

- Relevance and novelty: This paper focuses on a new problem for RL training (i.e. federated offline RL).

- Insightful discussion: The presentation of challenges related to federated offline RL is clear and insightful.

**Weaknesses:**

- The paper claims that it outperforms the centralized training offline RL, but the proposed method uses an additional data selection technique compared to the baselines, which is not quite fair. To make this claim, the baseline needs to also have data rebalancing or data selection technique (e.g. with [1], or just send the average reward of all episodes to the server).

- The title is "ensemble" method, but it seems the solution is not really using an ensemble method. It's a modified federated aggregation method.

- Even though the proposed method is about federated RL, related work and evaluation should be considered for supervised RL with heterogeneous data as well.

[1] Yue, Yang, et al. "Boosting offline reinforcement learning via data rebalancing." arXiv preprint arXiv:2210.09241 (2022).

**Questions:**

- The proposed technique seems to be doing a data selection based on the quality of the data, would you get a similar performance if you just select the clients with higher reward datasets? Like in [1] for example.

- Would the proposed method increase the amount of communication required for federated training compared to simple FAvg? It would be nice to quantify that.



[1] Yue, Yang, et al. "Boosting offline reinforcement learning via data rebalancing." arXiv preprint arXiv:2210.09241 (2022).

---

> ### Author Response · Authors · 2023-11-21
> **Response to Reviewer MJbF**
>
> We would like to thank the reviewer for their comments and suggestions. We are delighted to know that the reviewer finds  our paper well articulated, and the contents easy to follow. We are happy to know that the reviewer considers our problem framework novel. **In response to the reviewer's questions, we have conducted additional experiments and have added them to the appendix of our paper.** Below, we give detailed responses to  the specific questions, and we hope they address all the concerns and comments by the reviewer.
>
> **Q1** *``The paper claims that it ...the baseline needs to also have data rebalancing or data selection technique (e.g. with [1]( [Yue et al, 2022]), or just send the average reward of all episodes to the server)''*
>
> **Response:** Based on the reviewer’s comments, we have now added an additional baseline, TD3-BC with data rebalancing, which uses the the data rebalancing approach from [Yue et al, 2022], see Appendix E.1. We notice that data rebalancing does help, performing better than TD3-BC. However, its performance is still worse than our FEDORA algorithm. We believe this is due to the exponential weighting employed by FEDORA, and that data rebalancing methods do not completely solve the
> distribution shift issue that arises from combining data from multiple behavior policies. We would like to emphasize that the centralized offline RL algorithms serve only as a hypothetical baseline, because our setting federated learning without sharing the data.
>
> **Q2.**  *``The title is "ensemble" method ...''*
>
> **Response:** We believe that our approach bears a strong resemblance to ensemble learning in that we have a set of experts of heterogeneous qualities, whose collective wisdom we must combine together to obtain a high-quality policy. This is similar to ensemble learning, where weights must be determined for the different experts. Furthermore, we are constrained to evaluation only using local data. Hence, we also have an ensemble of critics, which must also be jointly combined to obtain an overarching critic. Hence, we named the algorithm as ensemble-directed to reflect its dependence the ensemble of actors and critics.
>
>
> **Q3.**  *``Even though the proposed method is about federated RL ...''*
>
> **Response:** We have now added references to two works ([Yue et al, 2022], [Yue et al, 2023]), thanks to the pointer by the reviewer in Q1. We will be happy to include more references if the reviewer has additional suggestions.
>
> **Q4.** *``The proposed technique ... would you get a similar performance if you just select the clients with higher reward datasets? Like in [1] ([Yue et al, 2022]) for example..''*
>
> **Response:** To answer this question, we have now **conducted two additional experiments (See Appendix E.2)**: $(i)$ we select clients based only on the rewards in their datasets using the weighing method proposed in [Yue et al, 2022], $(ii)$ we keep all the components of FEDORA such as optimistic critic, decaying the influence of local data, and the proximal term, but only modify the weighing mechanism, and use a similar method proposed in [Yue et al, 2022].
>
> The experiment results show that  selecting clients based only on the quality of data (i.e., method $(i)$) performs poorly, as there are several other relevant components that are needed  to ensure reasonable performance of federated RL with offline data. In method $(ii)$, we included all these additional components (optimistic critic, adding appropriate regulations, and decaying the influence of local data) but used the weighing mechanism suggested in [Yue et al, 2022]. The experiment results show that method $(ii)$ performs better than method $(i)$, but FEDORA still outperforms  it. This can be attributed to FEDORA's superior weighing strategy, and the holistic approach we took in designing it by augmenting different algorithmic components that help mitigate the problems of federated offline RL described in section 4.1.
>
> **Q5.**  *``Would the proposed method increase the amount of communication ..  compared to simple FAvg?''*
>
> **Response:** The communication required for  FEDORA  is similar to FedAvg. In FedAvg, during each round of federation, the subset of participating clients communicate  the  actor and critic parameters to the server. The server broadcasts the federated actor and critic parameters to all clients. In FEDORA, in addition to this, we only need to communicate a scalar weighing factor  which adds a negligible overhead.
>
>  [1] Yue, Yang, et al. "Boosting offline reinforcement learning via data rebalancing." arXiv preprint arXiv:2210.09241 (2022).

---

### Author Response · Authors · 2023-11-21
**General Response**

We extend our appreciation to the reviewers for their thoughtful insights and recommendations. We regret the delay in submitting our response, necessitated by the time required to conduct the additional experiments suggested by the reviewers. The main elements of our response are as follows:

1. In response to Reviewer MJbF's recommendation to compare FEDORA with centralized Offline RL algorithms employing data rebalancing, we have included an extra baseline in Appendix E.1. This new baseline is TD3-BC with data rebalancing [1].

2. In response to Reviewer MJbF's question on just select the clients with higher reward datasets, as done in [1], we conducted two additional experiments (see Appendix E.2). These experiments showcase FEDORA's superior performance and highlight our comprehensive approach in addressing various challenges encountered by federated offline RL algorithms.

3. Reviewers RUBc and mieC proposed experimenting with varying the temperature parameter $\beta$. We have incorporated these additional simulations in Appendix E.3

We emphasize that there is little work on federated offline reinforcement learning (as opposed to federated supervised learning), and ours is likely the first paper to illustrate why generic supervised federation is insufficient, while constructing an algorithm specifically for federated offline RL. We hope that all the additional experiments and our response will convince the reviewers the value of our work. We have provided detailed response to each reviewer's comments.

[1] Yue, Yang, et al. "Boosting offline reinforcement learning via data rebalancing." arXiv preprint arXiv:2210.09241 (2022).

---

### Meta-Review · Area_Chair_fPnV · 2023-12-06

**Metareview:**

This paper proposes a new federated offline reinforcement learning algorithm that addresses the setting with heterogeneous offline datasets, and learns a good policy without data-sharing. The setting of "offline RL" + "federated learning" is relatively new, though there have been a few works on the topic, e.g., Zhou et al., 2022 and some of its followups. The proposed approach has been tested on both simulated and real-world platforms. The paper is in general well-written and well-motivated, while some reviewers have concerns regarding the theoretical soundness, and the sufficiency of experimental results (e.g., the practicality of the real-world setting, and the comparison with sufficient baselines). Specifically, some design choices in the algorithms could have been better and more solidly justified -- e.g., it is known that the use of "optimistic" critic might not be a good idea in general in the offline RL setting (while Eq. 9 uses an optimistic one), what might be the effect of such a choice? Given the recent advances in offline RL, would it be possible to better characterize the sample efficiency of the algorithm under certain data assumptions? Regarding the practicality of the setting, a heterogeneous MDP setting might be more compatible and general than the single-MDP one, especially given the "multi-robot" real-world motivating and experimental setup of "cleaning robots in various houses". Addressing several aspects as mentioned above may help further improve the paper in its current form.

**Justification For Why Not Higher Score:**

The paper would have been made stronger if there were more theoretical justifications for the proposed approaches, more solid and complete experimental results. The writing of the paper may be improved also.

**Justification For Why Not Lower Score:**

N/A

---

### Decision · Program_Chairs · 2024-01-16

Reject